# Adrenal Failure: An Evidence-Based Diagnostic Approach

**DOI:** 10.3390/diagnostics13101812

**Published:** 2023-05-21

**Authors:** Salomi Shaikh, Lakshmi Nagendra, Shehla Shaikh, Joseph M. Pappachan

**Affiliations:** 1KGN Diabetes and Endocrine Centre, Mumbai 400001, India; 2Department of Endocrinology, JSS Medical College, JSS Academy of Higher Education and Research Center, Mysore 570015, India; 3Department of Endocrinology, Saifee Hospital, Mumbai 400004, India; 4Department of Endocrinology & Metabolism, Lancashire Teaching Hospitals NHS Trust, Preston PR2 9HT, UK; 5Faculty of Science, Manchester Metropolitan University, Manchester M15 6BH, UK; 6Faculty of Biology, Medicine and Health, The University of Manchester, Manchester M13 9PL, UK

**Keywords:** adrenal failure, cortisol, adrenocorticotrophin (ACTH), primary adrenal insufficiency (PAI), secondary adrenal insufficiency (SAI)

## Abstract

The diagnosis of adrenal insufficiency (AI) requires a high index of suspicion, detailed clinical assessment including detailed drug history, and appropriate laboratory evaluation. The clinical characteristics of adrenal insufficiency vary according to the cause, and the presentation may be myriad, e.g. insidious onset to a catastrophic adrenal crisis presenting with circulatory shock and coma. Secondary adrenal insufficiency (SAI) often presents with only glucocorticoid deficiency because aldosterone production, which is controlled by the renin angiotensin system, is usually intact, and rarely presents with an adrenal crisis. Measurements of the basal serum cortisol at 8 am (<140 nmol/L or 5 mcg/dL) coupled with adrenocorticotrophin (ACTH) remain the initial tests of choice. The cosyntropin stimulation (short synacthen) test is used for the confirmation of the diagnosis. Newer highly specific cortisol assays have reduced the cut-off points for cortisol in the diagnosis of AI. The salivary cortisol test is increasingly being used in conditions associated with abnormal cortisol binding globulin (CBG) levels such as pregnancy. Children and infants require lower doses of cosyntropin for testing. 21-hydoxylase antibodies are routinely evaluated to rule out autoimmunity, the absence of which would require secondary causes of adrenal insufficiency to be ruled out. Testing the hypothalamic–pituitary–adrenal (HPA) axis, imaging, and ruling out systemic causes are necessary for the diagnosis of AI. Cancer treatment with immune checkpoint inhibitors (ICI) is an emerging cause of both primary AI and SAI and requires close follow up. Several antibodies are being implicated, but more clarity is required. We update the diagnostic evaluation of AI in this evidence-based review.

## 1. Introduction 

Adrenal insufficiency (AI) is an uncommon but serious hormonal disorder which results from the complete or partial cessation of adrenal steroidogenesis to maintain normal health and well-being of the affected individual. AI occurs as a consequence of several diseases or insults to the adrenal glands and can present with very subtle symptomatology to catastrophic adrenal crisis that can even endanger life [1,2]. The signs and symptoms of AI vary widely and can mimic many other systemic disorders, delaying the clinical suspicion and diagnostic evaluation. Adrenal steroid synthesis can be affected by disorders involving the adrenal glands (primary AI), the diseases of the pituitary glands (secondary AI), or the hypothalamus (tertiary AI) with the reduced production of adrenocorticotrophin (ACTH) that stimulates adrenal steroidogenesis.

Regardless of the cause of AI, a prompt and timely clinical diagnosis is crucial for the optimal management of patients as the delay can lead to adrenal crisis during periods of stress, a condition associated with high morbidity and mortality [3,4,5,6]. Moreover, though classical cases of primary AI may not pose much diagnostic challenges to physicians and laboratory scientists, some patients with AI can present with atypical symptoms and signs and cause a dilemma in clinical and laboratory evaluation. Therefore, we attempt to appraise the latest evidence on the diagnostic evaluation of the disease to enable physicians to rationally approach any case of AI in this article.

## 2. Clinical Characteristics of Different Forms of AI

### 2.1. Primary Adrenal Insufficiency (PAI)

The clinical manifestations of PAI are influenced by the rate and degree of decline in the adrenal function. Clinical presentation may vary widely from vague, nonspecific symptoms of tiredness and fatigue to life-threating adrenal crisis where the patient may become unconscious with circulatory shock and hypothermia [4,7]. Weight loss, anorexia, nausea, vomiting, lethargy, salt craving, fatigue, muscle cramps, and postural hypotension are typically noticeable in primary adrenal failure. About 90% of individuals with chronic PAI have some level of skin hyperpigmentation, which is especially noticeable on the skin creases and scars, the extensor surface of the elbow, the knuckles, the lips, and the buccal mucosa. However, it should be emphasized that individuals with an acute onset of PAI, due to the destruction of the adrenal glands from infarction or hemorrhage resulting from acute vascular insults, would not have recognizable cutaneous hyperpigmentation [8]. Other characteristics of PAI include spontaneous episodes of hypoglycemia or an unexplained decrease in the insulin dose requirement in a person with diabetes mellitus, and in women, axillary and pubic hair loss owing to insufficient adrenal androgen production [9].

PAI may be inherited or acquired. The most common inherited form of PAI is congenital adrenal hyperplasia, while autoimmunity remains the primary cause of acquired PAI in the western world. However, various infections (bacterial, fungal, or mycobacterial) causing the destruction of the adrenal glands are the most common causes of PAI in the developing world [10,11]. Although PAI caused by autoimmune adrenalitis may occasionally occur in isolation without other endocrine dysfunction, more than 50% of autoimmune adrenalitis cases are associated with autoimmune polyglandular syndromes (APS) [12]. *Mycobacterium tuberculosis* (MTB) is the most common infection implicated in the development of PAI. Furthermore, infections with *Neisseria meningitides*, *Pseudomonas aeruginosa*, *Haemophilus influenzae*, *Pasteurella multocida*, *Staphylococcus aureus*, and Streptococci group A, when associated with septicemia, may result in an acute presentation of AI with bilateral adrenal infarction (known as Waterhouse–Friderichsen syndrome) often brought on by disseminated intravascular coagulation. Fever, nausea, myalgia, arthralgia, and an erythematous rash are the first symptoms, which proceed to cutaneous ecchymosis and purpura fulminans in the acute stage [13]. Other rare causes that can lead to the development of PAI include adrenal hemorrhage, metastatic cancer, infiltrative disorders, and certain drugs such as ketoconazole, fluconazole, etomidate, and mitotane. Drugs such as phenytoin and rifampicin increase the metabolism of cortisol and can precipitate adrenal insufficiency in susceptible individuals [14]. Patients with PAI usually show a pronounced impairment of aldosterone secretion from the total absence of adrenal steroidogenesis, leading to hyponatremia, hyperkalemia, and hypotension [15], as this mineralocorticoid hormone normally maintains the water electrolyte balance in the human body.

Adrenal crisis is an acute, often life-threatening condition that is more commonly associated with PAI when compared to cases of central AI [4]. The incidence of adrenal crisis is reported to be about 8% in patients with PAI [16]. The usual presentation of patients with adrenal crisis is dramatically diminished well-being, hypotension, nausea and vomiting, and fever that responds effectively to parenteral hydrocortisone treatment. Adrenal crisis is most often precipitated by infections. Other factors include hyperthyroidism, physical stress from surgery or trauma, inadvertent discontinuation of glucocorticoid medications, intense physical exercise, and psychological stress [17]. Early diagnosis, rehydration, and steroid replacement form the cornerstones of management, without which catastrophic outcomes including death can result.

### 2.2. Secondary Adrenal Insufficiency

The most prevalent subtype of adrenal insufficiency is secondary insufficiency, often resulting from a pituitary tumor [18]. A tumor of the hypothalamic-pituitary area is the most common endogenous cause of secondary adrenal insufficiency (SAI), and it is typically accompanied with panhypopituitarism due to pressure effects on the pituitary or the effects of surgery or radiation treatment for tumor removal. Less frequently, nontumoral diseases such as infiltrative lesions, hypophysitis, infectious processes, vascular changes, traumatic brain damage, empty sella syndrome, or genetic problems can also cause SAI [19]. Immune checkpoint inhibitor-induced SAI is increasingly being described in the recent literature, as these drugs are often effectively used for the treatment of various forms of cancers [3]. Clinical signs of SAI, in contrast to PAI, are only the result of glucocorticoid depletion from ATCH deficiency, because the production of aldosterone is often unaffected. Additionally, SAI seldom presents with an acute adrenal crisis, but an acute presentation may be brought on by stress from an injury, illness, or surgery. Symptoms and signs secondary to the deficiency of other anterior pituitary hormones as well as neurological signs can coexist, especially when the cause of SAI is a pituitary tumor [20].

### 2.3. Tertiary Adrenal Insufficiency

Tertiary adrenal insufficiency (TAI) is characterized by hypothalamic anomalies or dysfunction, which results in decreased corticotropin-releasing hormone (CRH) production, and consequently, the reduced synthesis of ACTH from the anterior pituitary gland. TAI, like SAI, is associated with diminished cortisol and dehydroepiandrosterone (DHEA) production from the adrenal glands because of a lack of ACTH stimulation. As a result, TAI is commonly referred to as central AI, or sometimes as SAI, because additional distinctive features between these two forms are sparse. TAI generally develops from the suppression of the hypothalamic–pituitary–adrenal (HPA) axis by the administration of supraphysiological doses of exogenous steroids over a prolonged period of time [21].

## 3. Laboratory Evaluation for Primary AI

### 3.1. 8:00 a.m. Cortisol

There should be a high index of suspicion for PAI in acutely unwell patients exhibiting clinical and biochemical features indicative of primary adrenal insufficiency (PAI), such as volume depletion, hypotension, hyponatremia, hyperkalemia, fever, abdominal discomfort, hyperpigmentation, or hypoglycemia (particularly in children). Treatment should not be delayed in patients who exhibit signs and symptoms of PAI, especially when they are acutely unwell, as delays may be associated with the development of catastrophic adrenal crisis. A baseline blood sample may be drawn for diagnostic testing and treatment with glucocorticoids, and prompt rehydration must be started without delay while awaiting test results [22].

Low morning cortisol concentrations (measured in serum or plasma before 8:00 a.m.) confirmed by low stimulated cortisol levels have typically been used to make the diagnosis of PAI. Though the optimal basal cortisol that can confirm or rule out adrenal insufficiency is a matter of debate, a morning serum cortisol of <140 nmol/L (5 μg/dL) in combination with increased ACTH levels (twice higher than the upper limit of normal laboratory values) are usually taken as confirmation of PAI [23]. A recent study demonstrated that a basal cortisol level of >12.83 μg/dL (354 nmol/L) usually rules out chronic PAI in an individual without stress factors such as infections [24] (see Figure 1). If only a basal cortisol test before 9:00 a.m. is performed, values of <100 nmol/L strongly suggest PAI [25].

Cortisol is predominantly transported in the body via cortisol-binding globulin (CBG) and albumin; only a small fraction (~10%) of the total serum cortisol is unbound and biologically active. Serum cortisol assays measure the total cortisol (bound and free forms), and their results can be misrepresentative in patients with altered serum protein concentrations [26]. Salivary cortisol was shown to correlate well with a serum-free cortisol level and was reported as a promising alternative test for the diagnosis of AI, especially among patients with altered CBG concentrations [27]. A basal morning salivary cortisol of <1.0 nmol/L may suggest PAI, while those with values of >5.9 nmol/L should have a confirmatory stimulation test [28].

### 3.2. Basal ACTH Measurement

When the cortisol production from the adrenal glands is reduced in individuals with PAI, their hypothalamic–pituitary–adrenal axis is activated by the stimulation of the hypothalamus and anterior pituitary from low plasma cortisol through the feedback loop (Figure 2). This results in an increased production of the corticotropin-releasing hormone (CRH) and ACTH. An ACTH level that is more than double the upper limit of the normal laboratory range in the presence of low basal plasma cortisol usually indicates PAI [29]. One needs to ensure that patients are not on corticosteroid therapy, including topical as well as inhaled forms of the drugs, before evaluating for PAI.

### 3.3. Stress Testing for Proving PAI

When the basal cortisol level is indeterminate (<354 nmol/L), a confirmatory test to prove AI is necessary. A short synacthen test (SST) or cosyntropin stimulation test are the most widely used stress testing methods to prove or disprove PAI. After securing the baseline plasma ACTH (if not tested earlier) and cortisol samples, 250 mcg of synacthen (cosyntropin; synthetic ACTH) is administered intravenously/intramuscularly with the collection of blood samples again at 30 and/or 60 min for estimating the plasma cortisol levels. A stimulated peek cortisol level of ≥500 nmol/L (18 μg/dL) excludes adrenal insufficiency as per the current international guidelines [23]. However, we have to bear in mind that when using newer highly specific cortisol assays such as liquid chromatography–tandem mass spectrometry (LC–MS/MS), the cut-off values can be lower at 412 and 485 nmol/L at 30 and 60 min (of stress testing), respectively, as shown by the newer studies [30].

As mentioned earlier, variations in the CBG levels can affect the usual cortisol assays, as these tests measure the total serum cortisol level. Therefore, in conditions with abnormal CBG levels such as pregnancy, nephrotic syndrome, liver cirrhosis, and estrogen therapy, the serum cortisol values may not be fully reliable to diagnose AI [31]. In such situations, a salivary cortisol assay can be used as a reliable alternative test. A very recent study demonstrated a sensitivity of 90.5% and specificity of 76.2% after SST for diagnosing AI using a stimulated salivary cortisol cut-off value of ≥16 nmol/L [32].

The gold standard dynamic test for assessing the entire HPA axis and adrenal failure is an insulin tolerance test (ITT). Hypoglycemia is one of the most powerful stressors for adrenal steroid secretion. The test is performed using intravenous insulin (0.1–0.15 unit/kg body weight) administration to induce hypoglycemia (plasma glucose <2.2 mmol/L), which results in a cortisol surge from the adrenal glands [29,33]. An ITT must be considered when the SST results are equivocal, or its diagnostic predictive value is doubtful. However, testing can be difficult in patients with cardiovascular disorders (such as ischemic heart disease) and neurological illnesses (contraindicated in patients with epilepsy as hypoglycemia of <2.2 mmol/L may precipitate a seizure episode). The overnight metyrapone stimulation test, which was used as an alternative to ITT in the past, is not often used in current clinical practice [29].

### 3.4. Other Laboratory Abnormalities in PAI

It is important to investigate the underlying cause of PAI for establishing the pathological diagnosis, as that has therapeutic and prognostic implications. As autoimmunity is the major cause of acquired PAI in the western world, 21-hydroxylase antibodies should ideally be checked in all cases [29]. If the patient tests positive for the antibodies, the patient should be further investigated for other autoimmune diseases such as type 1 diabetes mellitus, thyroid disease, premature ovarian failure, celiac disease, and atrophic gastritis with B12 deficiency, as PAI may be part of APS [34]. Organ-specific autoimmune disorders affecting other endocrine organs/body tissues may co-exist in patients with 21-hydroxylase auto-antibodies, or such autoimmunity can develop in these individuals in later life. Therefore, the current guideline of the Endocrine Society also recommends periodic testing for these conditions, ideally on an annual basis, if these tests are initially negative [23].

In the absence of adrenal autoantibodies, patients should be further investigated for structural defects in the glands such as infections, tumor infiltration, and bleed. A computed tomographic (CT) imaging is recommended, and other investigations such as testing for mycobacterial disease and fungal infections (in endemic areas) should be pursued according to the clinical need. Congenital disorders of adrenal steroidogenesis should be considered in patients younger than 20 years of age, and they should be worked up, as shown in Table 1. Molecular and genetic testing for these cases should help with planning definite management and prognostication.

PAI is usually associated with a concomitant deficiency of other adrenal steroids, and therefore, low aldosterone and DHEA levels are expected in most cases [29]. Hypoaldosteronism is associated with raised plasma renin activity, the measurement of which helps in monitoring the dose titration of mineralocorticoid replacement in such patients. Hyperkalemia and hyponatremia are manifestations of hypoaldosteronism classically seen in patients with PAI, whereas these abnormalities are uncommon in SAI and TAI (as aldosterone production is not just under the control of ACTH). Eosinophilia may be a common feature and a sensitive marker of AI [35].

### 3.5. Special Populations with PAI

The hormone profile may be different, and biochemical confounders in hormone assays may occur in some groups of patients such as pregnant women and children. Pregnancy is a physiological state of hypercortisolism, and the total cortisol level may be 2–3 times higher than normal in pregnant women due to increased levels of CBG as a consequence of stimulation from hyperestrogenemia, especially in the third trimester [29,36]. The cut-off levels for serum cortisol are not well-established in pregnant women, and therefore, the diagnostic evaluation for AI can be challenging both by basal hormone measurement and stress testing. Although the cut-off levels for pregnancy are not well-established, a salivary cortisol assay may be utilized for the diagnostic evaluation of AI in pregnancy [29,37].

In situations where gross alterations in the serum CBG levels can occur, as in patients with cirrhosis of liver, estrogen therapy, and nephrotic syndrome, an evaluation for AI with total plasma cortisol levels can pose diagnostic challenges. In such situations, both basal and stress-induced (e.g., SST) salivary cortisol measurements would provide effective alternate testing options [38]. In children and infants, lower doses of cosyntropin must be used for SST. In infants, cosyntropin 15 µg/kg body weight should be used, and children <2 years should be tested with 125 µg of cosyntropin [39].

## 4. Diagnostic Evaluation of SAI

### 4.1. Biochemical Tests

As SAI is usually a consequence of pituitary disease, abnormalities of other hormones from the gland may also co-exist in such patients. The degree of hormone abnormalities depends on the pathological alterations in the pituitary. For example, pituitary adenomas usually cause pressure-related damage to hormone-producing cells depending on the tumor size and rapidity of growth of the tumor. An overnight fasting static pituitary profile, ideally at 8:00 a.m., to estimate cortisol (+/− ACTH), gonadotropins coupled with gonadal hormones (estrogen in females and testosterone in males), prolactin, thyrotropin coupled with thyroxine, and insulin-like growth factor 1 (IGF-1 +/− growth hormone) levels should be performed for a baseline evaluation [40]. The hormone deficiency or excess state revealed by this baseline test will help further hormonal evaluation for proving the biochemical tests. Isolated ACTH deficiency causing SAI is usually a diagnosis of exclusion [29]. Provocative hormone tests to prove SAI can be performed using the similar methods discussed in the previous section on PAI. The ACTH stimulation test should be performed 6 weeks after surgery and at least 18 h after the last dose of hydrocortisone [40]. Individuals with SAI have lower z-scores for DHEAS, which has potential diagnostic value in younger patients and patients with tumors in assessing the integrity of the HPA axis, but more data is needed.

Pituitary insults causing SAI can also be the result of various systemic disorders such as hemochromatosis, sarcoidosis, immunoglobulin-4 (IgG-4) related systemic disease, and infectious diseases. Biochemical and immunological evaluations for such disorders should be performed as per the standard recommendations (not discussed here) for arriving at an appropriate diagnosis.

### 4.2. Imaging Studies

The standard diagnostic evaluation for pituitary disease following the biochemical diagnosis of SAI is a magnetic resonance imaging (MRI) of the gland. Screening pituitary via MRI prior to a biochemical diagnosis of SAI should be dissuaded, as the detection of an incidentaloma is likely in up to 10–40% of such imaging studies, [41] and unnecessary laboratory work-up and patient discomfort about the abnormal finding can be avoided. A review of the pituitary MRI by an expert neuroradiologist can give reasonable clue to a pathological diagnosis when a structural lesion is identified by such an imaging study in the presence of an appropriate pre-imaging biochemical diagnosis. Functioning and non-functioning pituitary adenomas, pituitary metastasis from primary malignancy at distal sites, hypophysitis, infections, and granulomatous disease can all result in SAI, and additional imaging studies such as CT imaging of the chest, abdomen, and pelvis, and positron emission tomography (PET) coupled with biochemical, immunological, and microbiological evaluation for such disorders should help in clinching the accurate diagnosis [42].

### 4.3. Evaluation for TAI

As TAI is usually a consequence of HPA axis suppression, a prompt clinical history should be obtained, including the use of topical and inhaled steroids; history of recent or current use of Indigenous medications; nutritional history including food supplements (and their contents); chronic administration of opioids; and self-administration of over-the-counter medications. The evaluation for various rare genetic syndromes associated with TAI (e.g., Prader–Willi syndrome) in patients with these syndromic features is indicated when appropriate. In addition, obtaining details about daily lifestyle including physical activities and various physical and emotional stressors are important, as these factors may be associated with marked alterations in the HPA axis [43].

### 4.4. Biochemical Work-Up

Hormonal evaluation usually involves the biochemical confirmation of AI, as described in the previous sections. A static pituitary profile with a normal or subnormal ACTH level with or without other pituitary hormone abnormalities may be present in patients with TAI. A lack of ACTH and cortisol response to ITT, and an increase in these hormone outputs following a CRH stimulation test, may point toward supra-hypophyseal disease causing TAI [44]. As the hypothalamus also controls the posterior pituitary hormonal balance, patients with TAI often have diabetes insipidus as a consequence.

### 4.5. Further Diagnostic Evaluation

Structural lesions including tumors and infiltrative lesions can often be diagnosed via MRI of the brainstem with a pituitary protocol to investigate patients with biochemically proven TAI. This should also aid in therapeutic decision making and prognostication [45]. Genetic studies are necessary in syndromic cases to prove the diagnosis, and for counselling purposes [46].

## 5. Immune Check Point Inhibitors (ICI) and AI

Cancer treatment has been revolutionized in recent years with the advent of ICIs. As these drugs can grossly modify the immune response in the human body, including that in the endocrine systems, the therapeutic use of many of these agents is associated with marked alterations in the body’s hormonal balance. Both the hyper- and hypofunction of several endocrine glands can occur with the use of ICIs [47]. Several agents of the ICI class have been used for cancer treatment in recent years, and many new molecules are under evaluation in clinical trials. Ipilimumab, pembrolizumab, nivolumab, atezolizumab, avelumab, durvalumab, and cemiplimab are some of the commonly used ICIs in current clinical practice. Different endocrine side effects varying from mild asymptomatic thyroid dysfunction to fatal adrenal crisis or diabetic ketoacidosis have been described with the use of this novel class of anticancer agents, and these endocrine toxicities may occur in 25–50% of patients [48].

Both PAI and SAI are common and can present after varying time intervals following the initiation of ICI therapy [47,48,49,50]. Though routine screening for asymptomatic cases is controversial, some centers screen patients for AI following every cycle of cancer treatment with ICIs [48]. PAI usually develops as a consequence of adrenalitis, whereas SAI can develop from hypophysitis or isolated ACTH deficiency [49]. Several autoantibodies have been characterized in patients with ICI-induced PAI and SAI [50]. Those associated with PAI are anti-21-hydroxylase, anti-17α-hydroxylase, anti-P450 side chain cleavage enzyme, anti-aromatic L-amino acid decarboxylase (AADC), anti-interferon (IFN)α, and anti-IFNΩ antibodies. Antibodies associated with SAI include anti-guanine nucleotide-binding protein G(olf) subunit alpha (GNAL), anti-pro-opiomelanocortin (POMC), anti-TPIT (corticotroph-specific transcription factor), anti-integral membrane protein 2B (ITM2B), anti-zinc finger CCHC-type containing 8 (ZCCHC8), and anti-pituitary-specific transcriptional factor-1 (PIT-1). However, the utility of these antibodies for routine clinical evaluation and screening are not yet clear.

Several endocrine societies have developed guidelines for the diagnosis and management of ICI-related endocrine dysfunction (beyond the scope of this review) and can be utilized as per the demand of the clinical situation [51,52,53,54].

## 6. Areas of Uncertainty

Although several clinical practice guidelines elaborate the diagnostic approach to various types of adrenal failure, there is no clear consensus on the cut-off levels for basal and stimulated cortisol in special circumstances where CBG levels are altered (e.g., pregnancy, estrogen therapy, cirrhosis, and nephrotic syndrome). There is still uncertainty about the exact cut-off levels for the suggested alternative test, salivary cortisol (basal and stimulated) test, in these situations, as there is only meagre evidence of its clinical utility. Similarly, testing algorithms and the utility of various antibodies for diagnostic evaluation of patients with ICI-induced AI is still evolving, especially because newer molecules are rapidly being added to this group of anticancer agents.

## 7. Conclusions

Adrenal failure is an uncommon but life-threatening endocrinopathy with protean manifestations. The diagnosis can often be missed without prompt clinical evaluation and diagnostic work-up and may lead to devastating complications in the form of lethal adrenal crisis. The clinical profile and laboratory work-up for the disease vary according to the type of insult, namely, PAI, SAI, or TAI. Prompt assessment of the basal and stimulated cortisol level followed by antibody testing, testing for the integrity HPA axis, various imaging techniques, and testing for systemic insults as causes of AI are the usual approaches to diagnostic evaluation. Following various international clinical guidelines, when appropriate, also helps us to plan appropriate laboratory evaluations of this uncommon but important endocrine disorder.

## Figures and Tables

**Figure 1 diagnostics-13-01812-f001:**
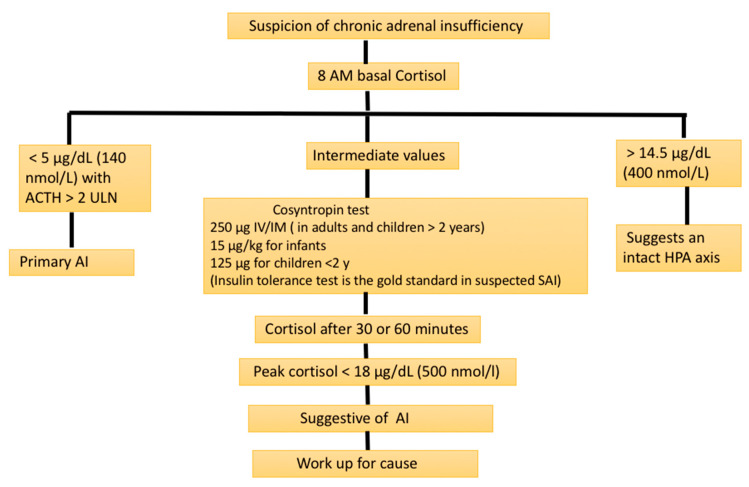
Biochemical screening evaluation for suspected adrenal insufficiency (AI). ACTH: adrenocorticotrophin; HPA axis: hypothalamic–pituitary–adrenal axis; SAI: secondary adrenal insufficiency; ULN: upper limit of normal.

**Figure 2 diagnostics-13-01812-f002:**
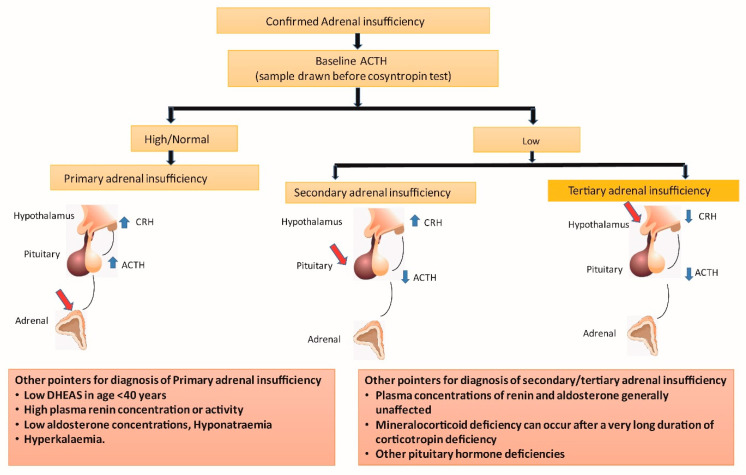
Alterations in hypothalamic–pituitary–adrenal axis in various forms of adrenal insufficiency. ACTH: adrenocorticotrophin; CRH: corticotrophin-releasing hormone; DHEAS: dehydroepiandrosterone sulphate; Red arrows indicate site of disease; Blue arrows indicate high (upward) or low (downward) hormone levels.

**Table 1 diagnostics-13-01812-t001:** Various forms of adrenal insufficiency, the causes, and diagnostic evaluation. AI: adrenal insufficiency; APS: autoimmune polyglandular syndrome; APLA: antiphospholipid antibody; CAH: congenital adrenal hyperplasia; CHF: congestive heart failure, CMV: cytomegalovirus; CRH: corticotrophin-releasing hormone; CT: computed tomography; MRI: magnetic resonance imaging; VLCFA: very long chain fatty acid; * after ruling out pheochromocytoma.

Disease Entity	Cause	Diagnostic Evaluation
*Primary AI*		*High baseline ACTH, low basal & stimulated serum cortisol*
Autoimmune AI Isolated autoimmune adrenalitis Associated with APS-I Associated with APS-II Infection	Immune reaction against 21-hydroxylase	Autoantibodies against 21-hydroxylase
Tuberculosis HIV	Mycobacterium tuberculosis Opportunistic infections/neoplasms	CT Antigen detection tests/blood cultures
Fungal	Histoplasmosis Paracoccidioidomycosis	
Viral	CMV	
Bilateral adrenal hemorrhage	Sepsis CHF APLA Use of anticoagulants Trauma Surgery	CT
Adrenal metastasis		
	Malignancy	CT/biopsy *
Adrenal infiltration		
	Amyloidosis Sarcoidosis Lymphoma	CT/biopsy *
Genetic AI	X-linked adrenoleukodystrophy CAH Adrenal hypoplasia congenita ACTH insensitivity syndrome	VLCFA 17-OHP NR0B1 mutation MC2R mutation
Drug induced	Various adrenolytic agents	Prompt medication history
*Secondary AI*		*Low basal ACTH, low stimulated serum cortisol*
Space occupying lesions Trauma Surgery/irradiation	Pituitary tumors Altered blood supply	MRI MRI MRI
Infection	Tubercular hypophysitis Pyogenic abscess	MRI/biopsy
Infiltration	Lymphocytic hypophysitis Sarcoidosis Wegener’s granulomatosis Histiocytosis X	MRI/biopsy
Sheehan’s	Severe blood loss/hypotension causing necrosis of the pituitary	MRI
Drugs	Ketoconazole Fluconazole Etomidate Metyrapone	
*Tertiary AI*		*Low basal ACTH, low stimulated cortisol* *Increased ACTH and cortisol response to CRH stimulation test*
Space occupying lesions Trauma Surgery Irradiation Drug induced	Hypothalamic tumors Glucocorticoids	MRI

## Data Availability

Not applicable.

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
