# Peer review of "Adrenal Failure: An Evidence-Based Diagnostic Approach"

_diagnostics, 2023, doi:10.3390/diagnostics13101812_

Round 1
Reviewer 1 Report
In this paper, the authors present the complexity of the diagnosis of adrenal insufficiency. The subject is of great interest and can be seen as a lead to a new approach in the diagnosis and management of adrenal failure.
The introduction focuses on the general knowledge on the subject. The information provided is useful in understanding the topic and the importance of the subject in nowadays medicine.
The methods used are complex and well described.
The authors conducted a proper collection of the data. The information presented is up to date, suitable, and substantial.
The conclusions are coherent and sustain the findings.
The figures and tables presented are easy to interpret and understand.
Good English level.
I recommend it for publication.
Author Response
Reviewer 1
In this paper, the authors present the complexity of the diagnosis of adrenal insufficiency. The subject is of great interest and can be seen as a lead to a new approach in the diagnosis and management of adrenal failure.
The introduction focuses on the general knowledge on the subject. The information provided is useful in understanding the topic and the importance of the subject in nowadays medicine.
The methods used are complex and well described.
The authors conducted a proper collection of the data. The information presented is up to date, suitable, and substantial.
The conclusions are coherent and sustain the findings.
The figures and tables presented are easy to interpret and understand.
Good English level.
I recommend it for publication.
Author responses:
Thanks for the encouraging comments from the expert reviewer. We also thankfully acknowledge the effort in reading the paper and make recommendations as above.
Reviewer 2 Report
The manuscript submitted by Dr. Shaikh et al. presents diagnostic issues connected with adrenal insufficiency. The subject is of interest, however, there is not much novelty in this area. Nonetheless, a comprehensive review of the diagnostics procedures in adrenal failure might be useful for the sake of education. There are just a few minor points that could be better explained in the text.
When writing about factors which may precipitate adrenal crisis it is worth mentioning hyperthyroidism, which frequently co-exists with autoimmune PAI.
In SAI lack of ACTH may actually affect adrenal androgen synthesis, because early steroidogenesis enzymes, common for glucocorticoid and androgen pathway, remain under ACTH control. Furthermore, in pituitary disorders overall androgen level can be additionally decreased due to lack of proper gonadotropin stimulation. Nonetheless, signs and symptoms of glucocorticoid depletion are vitally much more important, than lack of androgens.
Emphasis on testing for additional autoimmune conditions is particularly noteworthy, as it has clear clinical implications – accompanying diseases may seriously alter the metabolic balance, or even turn out life-threatening in some instances.
In the sentence: “Pituitary insults causing SAI can often be the result of various systemic disorders such as haemochromatosis, sarcoidosis, immunoglobulin-4 related disease, and infectious diseases.” I would rather consider that the above conditions are relatively rare, uncommon causes of hypopituitarism.
Description of the immune check point inhibitors is of special interest with regard to “modern” reasons for adrenal failure, primary as well as secondary.
To summarize, although not particularly novel, the paper seems a reliable summary of the current knowledge about adrenal failure diagnostics and could be published in Diagnostics.
Minor English revision is required, mostly typos.
Author Response
The manuscript submitted by Dr. Shaikh et al. presents diagnostic issues connected with adrenal insufficiency. The subject is of interest, however, there is not much novelty in this area. Nonetheless, a comprehensive review of the diagnostics procedures in adrenal failure might be useful for the sake of education. There are just a few minor points that could be better explained in the text.
When writing about factors which may precipitate adrenal crisis it is worth mentioning hyperthyroidism, which frequently co-exists with autoimmune PAI.
In SAI lack of ACTH may actually affect adrenal androgen synthesis, because early steroidogenesis enzymes, common for glucocorticoid and androgen pathway, remain under ACTH control. Furthermore, in pituitary disorders overall androgen level can be additionally decreased due to lack of proper gonadotropin stimulation. Nonetheless, signs and symptoms of glucocorticoid depletion are vitally much more important, than lack of androgens.
Emphasis on testing for additional autoimmune conditions is particularly noteworthy, as it has clear clinical implications – accompanying diseases may seriously alter the metabolic balance, or even turn out life-threatening in some instances.
In the sentence: “Pituitary insults causing SAI can often be the result of various systemic disorders such as haemochromatosis, sarcoidosis, immunoglobulin-4 related disease, and infectious diseases.” I would rather consider that the above conditions are relatively rare, uncommon causes of hypopituitarism.
Description of the immune check point inhibitors is of special interest with regard to “modern” reasons for adrenal failure, primary as well as secondary.
To summarize, although not particularly novel, the paper seems a reliable summary of the current knowledge about adrenal failure diagnostics and could be published in Diagnostics.
Author responses:
Thanks for the encouraging comments from the expert reviewer.
We agree that there is nothing novel in this review and we acknowledge that we summarise the current evidence on the topic.
We have made minor changes in the relevant sections of the revision to incorporate the suggestions of the reviewer.